# Efficacy of Household and Commercial Washing Agents in Removing the Pesticide Thiabendazole Residues from Fruits

**DOI:** 10.3390/foods14020318

**Published:** 2025-01-18

**Authors:** Xinyi Du, Lauren Ho, Sisheng Li, Jeffery Doherty, Junghak Lee, John M. Clark, Lili He

**Affiliations:** 1Department of Food Science, University of Massachusetts, Amherst, MA 01003, USA; xinyidu@umass.edu (X.D.); jjd@umass.edu (J.D.); 2Department of Veterinary and Animal Sciences, University of Massachusetts, Amherst, MA 01003, USAjclark@vasci.umass.edu (J.M.C.); 3Department of Chemistry, University of Massachusetts, Amherst, MA 01003, USA

**Keywords:** pesticide residue removal, postharvest washing strategies, SERS, thiabendazole, non-ionic surfactants, LC-MS/MS

## Abstract

Pesticide residues on fruits pose a global food safety concern, emphasizing the need for effective and practical removal strategies to ensure safe consumption. This study investigates the efficacy of household ingredients (corn starch, all-purpose flour, rice flour and baking soda) and four commercial fresh produce wash products in eliminating a model pesticide thiabendazole with and without a model non-ionic surfactant Alligare 90^®^ from postharvest fruits. Surface-enhanced Raman spectroscopy (SERS) was employed for the rapid, in situ quantification of residue removal on apple surfaces. Soaking in 2% corn starch followed by soaking in 5% baking was the most effective homemade strategy, removing 94.13% and 91.78% of thiabendazole with and without the surfactant. Among commercial washing agents, soaking in 2% Product 4 demonstrated the highest efficiency, removing 95.3% and 95.99% of thiabendazole with and without surfactant. These results suggested that the non-ionic surfactant did not affect removal efficiency. Both protocols were effective across various fruits (apples, grapes, lemons, strawberries), validated by liquid chromatography–mass spectrometry (LC-MS/MS) analyses. However, safety concerns regarding the composition of Product 4 highlighted the benefits of homemade strategies. Overall, this work offers practical guidelines for reducing pesticide residues on fruits and enhancing food.

## 1. Introduction

Pesticides were extensively applied in agriculture for crop protection, but the health risks associated with their residues on food were unignorable [1,2,3]. The 2021 European Union report revealed that, among 87,863 tested food samples, 44.3% contained detectable levels of one or more pesticides, with fruits such as apples, grapes, and strawberries exhibiting the highest incidence of multiple pesticide residues in unprocessed food [4]. With the rising consumption of fruits, it is crucial to develop effective methods to minimize pesticide residues on fruits to safeguard consumer health.

Washing is one of the most common and convenient operations for eliminating pesticide contaminants on both home and industrial scales, causing minimal impact on fruit quality compared to other processing methods, like drying, freezing, peeling, and cooking [5,6,7]. As the majority of pesticides accumulate on the peel surface of fruits [8,9], washing can significantly diminish their residue levels. However, washing with tap water, which removes residues by dissolving pesticide active ingredients (AIs) in water, demonstrates limited efficacy for removing lipophilic AIs [7,10,11]. The use of non-toxic household substances, such as baking soda (sodium bicarbonate), acetic acid, and salt (sodium chloride), has emerged as a sustainable and practical approach for removing pesticides without harming the fruits [9,12,13]. Notably, baking soda has been identified as an especially effective agent [14,15,16], attributed to its role in the alkaline hydrolysis of pesticides and the cuticular wax layer [12,17,18]. Furthermore, the potential of starches for contaminant adsorption has been increasingly recognized [12,19,20]. Despite the proliferation of commercial produce wash products claiming efficacy in removing pesticide residues, the U.S. Food and Drug Administration (FDA) has not officially recommended these products, as their effectiveness has not been comprehensively tested [21]. Additionally, the porous nature of fruits allows them to absorb some substances, which may not be completely removed even with thorough rinsing, posing a new health risk [21]. Therefore, there is an urgent need for research into safe, effective and practical washing strategies that utilize ingredients generally recognized as safe (GRAS), enabling consumers to apply them at home easily.

The objective of this study is to develop effective washing strategies using household or commercial agents to remove pesticide residues from fruits. Thiabendazole, a systemic fungicide approved for postharvest application on various fruits, was selected as the model pesticide. Although thiabendazole is generally considered low in acute toxicity to mammals (oral LD50 for rats: 2080 mg/kg) [22], it poses potential developmental toxicity risks [23]. Furthermore, its persistence on fresh produce has raised significant food safety concerns [24,25]. The maximum residue limit (MRL) of thiabendazole set by the Environmental Protection Agency (EPA) is 10 µg/g, 0.05 µg/g, 10 µg/g, and 5 µg/g for apple, grape, lemon, and strawberry, respectively [26]. Alligare 90^®^, a commercial non-ionic surfactant containing 20~30% alkylphenol ethoxylates (APEOs) [27], was chosen as the model adjuvant. The inclusion of a pesticide active ingredient with adjuvant, representing a more complex but practical scenario, has been less explored.

This study evaluated the reduction in thiabendazole residues on apples using SERS-based methods. Surface-enhanced Raman spectroscopy (SERS) is a spectroscopic technique that provides molecular “fingerprints” to identify analytes adsorbed on or in the vicinity of SERS-active substrate surfaces through the collection of molecular vibrations [28,29,30,31]. It stands out for its simplicity and efficiency over traditional chromatography methods like GC-MS/MS or LC-MS/MS, which require time-consuming extraction and cleanup processes, well-trained personnel and expensive equipment [32,33,34,35]. SERS has proven effective for the in situ detection of low levels of pesticides on a variety of fresh produce. For example, Hou et al. [36] identified four different insecticides on fresh tea leaves with limits of detection (LODs) ranging from 0.25 to 0.5 mg/kg, Ma et al. [37] quantitatively analyzed chlorpyrifos in tomatoes down to 0.001 mg/L (or 2.85 × 10^−9^ mol/L), and Wang et al. [38] monitored thiram in herbal plants with a detection limit of 8.7 × 10^−9^ mol/L. Our prior studies reported that the SERS-based method coupling with a gold nanoparticle (AuNP) mirror substrate was effective at detecting pesticides in the presence of non-ionic surfactants in postharvest produce [18,39].

The efficiency of soaking using household agents, starch (all-purpose flour, corn starch and rice flour), and baking soda were assessed, and, then, the method was further optimized by extending soaking time or following sequential soaking. Additionally, the efficacy of four commercial fresh produce wash products was tested, according to their instructions and an optimized soaking protocol, to establish another effective washing strategy using the commercial agent. Furthermore, the pesticide removal efficiencies of the developed washing strategies were further validated in different fruit models (apple, grape, lemon, and strawberry) using the LC-MS/MS method. To our knowledge, this is the first comprehensive study to assess the cleaning effect of various household agents and commercial agents on different fruit matrices. This research offers some practical guidelines for consumers to eliminate pesticide residues from fresh produce at home.

## 2. Materials and Methods

### 2.1. Materials

Thiabendazole (4-(1H-1,3-Benzodiazol-2-yl)-1,3-thiazole) of analytical grade (≤100%) was purchased from Sigma-Aldrich (St Louis, MO, USA). Alligare 90^®^ was acquired from Alligare, L.L.C. (Opelika, AL, USA). The organic solvent methanol (≥99.8%), acetonitrile (≥99.8%), and hexane (≥95%) were obtained from Fisher Scientific (Fair Lawn, NJ, USA). A 50 mg/L solution of citrate-capped NanoXact Gold Nanospheres (particle size: 50 nm) was purchased from nanoComposix (San Diego, CA, USA). All-purpose flour (Gold Medal^®^), corn starch (Argo^®^), white rice flour (Anthony’s^®^), baking soda powder (Arm & Hammer^®^), and four different Fruit and Vegetable Wash products were procured from online sources. Four types of organic fruits, including ‘Royal’ gala apple, green grape, lemon, and strawberry were obtained from Whole Foods (Amherst, MA, USA).

### 2.2. Fruit Sample Preparation

Thiabendazole powder was first dissolved in methanol to prepare the stock solutions (1000 mg/L), while the 1% (*v*/*v*) Alligare 90^®^ solution was diluted with ultrapure water for further use. The solution of thiabendazole alone (T) (AI: 100 mg/L) was diluted from a thiabendazole stock solution with ultrapure water, while the corresponding formulation of thiabendazole with Alligare 90^®^ (TA) (AI: 139 mg/L) was prepared using the thiabendazole stock solution and 1% Alligare 90^®^ dilution to attain a final Alligare 90^®^ concentration of 0.5%. Notably, the AI concentration was increased by 39% in the presence of 0.5% Alligare 90^®^ to adjust for the increased wetted area of each droplet, as measured in the prior study [39]. Water suspensions or solutions of corn starch (1% and 2%), all-purpose flour (2%), rice flour (2%), baking soda (5%), and a mixture of corn starch (2%) and baking soda (5%) were freshly prepared by stirring (1000 rpm, 5 min) as homemade soaking agents. Four commercial washing agents were utilized by two methods: (1) following their instructions (soaking or spraying), as shown in Appendix A, and (2) preparing 2% soaking solutions.

The organic gala apple was chosen as the model fruit for evaluating thiabendazole residue removal after different washing treatments using the SERS technique. Multiple aliquots with a volume of 5 μL of thiabendazole-based formulations (T or TA) were directly pipetted onto a clean and dry apple at room temperature (22 ± 2 °C). After drying in a fume hood for 1 h, the apple fruit was either immersed in a 200 mL soaking solution for a predetermined duration or treated following the instruction of the selected commercial agents and then gently rinsed under running deionized water for 30 sec. After air-drying at room temperature, apple peels (approximately 3 mm thickness) were peeled from the whole apple for further pesticide analysis using SERS. Based on the SERS analysis results, two of the most effective washing strategies for cleaning thiabendazole residues were obtained. TA was applied on apples, grapes, lemons, and strawberries, achieving a final thiabendazole concentration of 20 µg/g for each. The residue values in various fresh produce models after employing the two washing strategies were quantified using the LC-MS/MS method, respectively. It is important to note that, to minimize variations caused by mechanical forces, the washing process involved only immersing and rinsing without any scrubbing.

### 2.3. Characterize the Properties of Starch and Starch Suspension

Starch granules were observed in bright-field mode by using a DXR3xi Raman imaging microscope (Thermo Fisher Scientific, Madison, WI, USA) controlled by OMNIC™ software (version 9.1) The samples were prepared by sprinkling a small amount of starch powder on a clean and dry microscope slide. The size of starch particles was measured using ImageJ software (version 1.54).

The stabilities of the selected starch suspensions were analyzed by Turbiscan™ LAB (Formulaction, Toulouse, France). The Turbiscan Stability Index (TSI) was measured every 30 s to characterize the physical stability of each suspension over time.

The distribution of thiabendazole residues in starch suspension, removed from apples treated with thiabendazole-based formulations, was monitored using a confocal laser scanning microscope (CLSM, Nikon Eclipse C1 80i, Nikon Instruments Inc., New York, NY, USA). Apples untreated with pesticide were washed to make the control group. Each sample was prepared by placing one droplet on a microscopic glass slide with cover glass for observation under microscope.

The zeta potentials of starch suspensions were measured using an electrophoresis instrument (Zetasizer Nano ZS, Malvern Instruments, Malvern, UK).

### 2.4. Fabrication and Utilization of SERS-Active Substrate

Mirror-like SERS-active substrates were prepared following methods previously reported [40,41]. Briefly, 5 mL of acetonitrile (a relative polar solvent) was mixed with 5 mL of hexane (a non-polar solvent) by vortexing. This process produced two discernible solvent layers, of which the bottom polar layer was collected and reserved for future use. Subsequently, 100 μL of the separated bottom solvent layer was gradually introduced to 50 μL of AuNPs (250 mg/L), which were concentrated from 50 mg/L of commercial citrate-capped AuNPs by centrifugation, to form a mirror-like AuNP substrate pellet. These AuNP mirror substrates were then transferred using a pipette to the areas where pesticide droplets had been previously applied on apples. Following air-drying at room temperature, SERS mapping images were collected from the AuNP mirror-covering area on the apple surfaces.

### 2.5. Quantitative Analysis of Thiabendazole Residue Using SERS

A DXR Raman microscope (Thermo Fisher Scientific, Madison, WI, USA) equipped with a 780 nm laser and a 20× objective lens was used in this study. The laser power was set at 5 mW, and the exposure time for signal collection was set at 2 s. The Raman spectra were collected from 2000 cm^−1^ to 400 cm^−1^ using a 50 μm slit aperture. Each SERS mapping image (200 μm × 200 μm) collected from the mirror-covered area comprised 25 spectra as the step size set as 50 μm. Signals from different mappings in each scenario were averaged and further analyzed using OMNIC™ software (version 9.1) and TQ Analyst software (version 8.0).

For the quantitative analysis of thiabendazole residues on apples, ten-point calibration curves of T (concentration: 0–1000 mg/L) and TA (concentration: 0–1390 mg/L) were constructed and modified from a prior study [18]. The working standards of T were prepared by serial dilution using ultrapure water, while the working standards of TA were diluted using 1% Alligare 90^®^ and ultrapure water to achieve a final concentration of Alligare 90^®^ concentration of 0.5%. Four aliquots with a volume of 5 μL of each working standard were deposited on the clean apple peels and air-dried for 20 min. After being covered with AuNP mirrors, SERS signals from each replicate of the working standards were collected. Calibration curves for T and TA were constructed based on the characteristic peak height at 1010 cm^−1^ versus mass/area (ng/mm^2^), which was determined by dividing the thiabendazole mass in a 5 µL droplet by the observed wetted area of each droplet of pesticide solution on an apple (6.126 mm^2^ for T). Calibration curve parameters were determined through linear and stepwise fitting using OriginPro software (version 2023b). The characteristic SERS peak heights at 1010 cm^−1^ for the washed groups were utilized to estimate the residue levels according to the quantitative curves.

### 2.6. Validation of the Removal Efficiency of Washing Strategies Using LC-MS/MS

The amounts of internal pesticide residues (µg/g) in thiabendazole-treated apples, grapes, lemons, and strawberries following the most effective washing strategies were quantified using LC-MS/MS. The internal pesticides were first extracted by a QuEChERS standard operating procedure [42]. Residues in each extraction were analyzed using a Waters Alliance LC system coupled with a Waters Acquity TQD MS/MS system at the Massachusetts Pesticide Analysis Laboratory (Amherst, MA, USA). The chromatographic separation was carried out on an Atlantis T3 column (2.1 × 100 mm; 3 µm, Waters^TM^, Milford, MA, USA), maintained at 30 °C. The mobile phase consisted of 0.1% formic acid, 5 mM of ammonium formate in water (Phase A), and 0.1% formic acid, 5 mM of ammonium formate in acetonitrile (Phase B). The initial condition of 95:5 A/B was held for 0.5 min, followed by a gradient increasing Phase B to 50% over 0.5 min and increasing Phase B to 100% at 4 min, sustained until 10.5 min, and then increasing Phase A to 95% at 11 min, with an equilibration time of 4 min. The flow rate was kept at 0.2 mL/min, and the injection volume was set 10 µL. The capillary voltage was 3000 V, and high-purity argon (99.999%) was used as collision gas. The ion source temperature was at 150 °C, and nitrogen was utilized for desolvation. Chromatograms equipped with a positive electrospray ionization (ESI+) source and operated in Multiple Reaction Monitoring (MRM) mode included the following settings: collision gas, 0.2 mL/min; retention time, 5.22 min; precursor ion, 201.96; quantifying ion, 65; qualifying ion, 175.

### 2.7. Statistical Analysis

Significant differences in the analysis results of various washing strategies, analyzed using SERS and the LC-MS/MS method, were determined by one-way analysis of variance (ANOVA) with Bonferroni’s test at a confidence level of 95% using OriginPro software.

## 3. Results and Discussion

### 3.1. Characterization and Quantitative Analysis of Systemic Thiabendazole Residue on Apple Surfaces

The SERS spectra and the quantitative analysis method for thiabendazole residues on apples, both with and without Alligare 90^®^, using AuNP mirrors were reported in our previous study [18,43]. Figure 1A reveals that the dominant characteristic SERS peak for both T and TA were observed at 785, 1010, 1275, and 1543 cm^−1^. While the peak at 1543 cm^−1^ was attributed to C=N stretching, the remaining peaks were associated with C-H vibration modes [44,45,46]. The peak around 1010 cm^−1^ was chosen as the quantitative analysis benchmark for thiabendazole residues in the absence and presence of Alligare 90^®^ in subsequent studies.

To quantify thiabendazole residues on apple surfaces, the mass/area-dependent calibration curves of T and TA were established based on the Raman peak height at 1010 cm^−1^. SERS signals for both pesticide formulations were collected at ten different concentrations, expressed as mass/area ranging from 0 to 816.25 ng/mm^2^. The prior study demonstrated a notable difference in the y-intercepts, but not in the slopes, of the linear regression curves for T and TA within the mass/area range of 0–4.08 ng/mm^2^ (Appendix A) [18]. The limits of detection (LOD) and quantification (LOQ) were measured as 0.745 ng/mm^2^ and 2.256 ng/mm^2^ for T, and 0.153 ng/mm^2^ and 0.460 ng/mm^2^ for TA, respectively. These results suggested an enhanced sensitivity of the SERS detection method for thiabendazole in the presence of Alligare 90^®^, which was attributed to APEOs promoting the transport of thiabendazole through the apples’ wax layer, thereby facilitating the interaction between pesticide and AuNPs [39]. Different from our earlier work, two additional working standards (408.12 and 816.25 ng/mm^2^) were included in each quantitative curve to account for high pesticide concentration levels. The stepwise regression curves of T and TA were constructed within the mass/area range of 4.08–816.25 ng/mm^2^ as shown in Figure 1B and Table 1. A nonlinear positive correlation was observed up to 105.23 ng/mm^2^ for T and 82.15 ng/mm^2^ for TA. Beyond these ranges, the characteristic SERS peak height at 1010 cm^−1^ was expected to keep in constant values of 557.25 for T and 590.28 for TA, indicating that residue value calculation based on the quantitative curves is impracticable when signal intensities exceed these constants. The thiabendazole residues in the absence and presence of Alligare 90^®^ on apple surfaces were determined based on the corresponding quantitative curves of T and TA in the subsequent study.

### 3.2. Optimizing the Starch-Based Soaking Strategy

The particle size and morphology of all-purpose flour, corn starch, and rice flour were revealed by high-resolution optical microscopy images in Figure 2A. Corn starch granules exhibited a spherical shape with smooth edges and a uniform particle size within 3–20 µm in diameter, while white rice flour particles were polygonal with rough edges, displaying a wide particle size distribution from nanometers to over 200 µm. All-purpose flour consisted of both smooth spherical granules and rough polygonal particles, with a diameter range of 1–50 µm.

Apples applied with a 1000 ppm thiabendazole solution (AI mass/area: 816.25 ng/mm^2^) were submerged in three different starch suspensions (2%) and water for 10 min. To be noted, all the starch suspensions were freshly prepared before being used for soaking due to the quick formation of sedimentation. The thiabendazole residue value (mass/area) following each washing treatment was quantified using the SERS mapping method, based on the calibration equations (Table 1). The SERS analysis results, including the Raman peak height at 1010 cm^−1^, corresponding residue values (mass/area), and reduction in thiabendazole (%) are summarized in Appendix A. Soaking with pure water demonstrated limited efficacy, with some replicates showing residue levels exceeding the calibration range (Raman peak intensity > 557.25). Thus, Figure 2B directly compared the residue signal intensities following soaking with different suspensions, indicating that starch suspensions facilitated more effective thiabendazole removal from apples than pure water. Among the starches tested, corn starch proved to be the most effective agent by reducing the thiabendazole residues to 24.69 ± 5.26 ng/mm^2^, while rice flour was the least effective agent. The effectiveness of the starches (corn starch > all-purpose flour > rice starch) appeared to inversely correlate with their particle size and roughness. Lian et al. [19] reported similar results that the smaller starch particles had higher removal efficiency of lambda-cyhalothrin from lettuce. Hence, the superior performance of corn starch is likely attributed to its small particle size, large surface area, and porosity [47], which enable the efficient physical absorption of pesticides through van der Waals forces. Furthermore, starch is rich in hydroxyl groups [48], which can form hydrogen bonds with the thiabendazole’s amino groups. Therefore, unmodified starch, especially corn starch, can remove thiabendazole from apple surfaces effectively via their physisorption and chemisorption behavior in the aqueous system.

Lian et al. [19] demonstrated that starch granules, as interfacial stabilizers, could remove pesticide residues by facilitating the formation of O/W Pickering emulsions. However, the emulsifying capability of native starches was challenging due to their large particle size [49,50]. In addition, previous experiments validated the emulsifying capability of starches employing organic solvents instead of pesticide AIs to simulate O/W formation [19,49,50,51], which is different from actual fruit washing practices. This study observed the thiabendazole removed from the apples using a corn starch suspension, without homogenization, under optical microscopy. No formation of the O/W Pickering emulsion was observed in Figure 3, which was likely due to the large particle size of starch granules used in this study, and the low content and partition coefficient (log K_ow_ = 2.47) [52] of thiabendazole residues. Therefore, the cleaning mechanisms are physisorption and chemisorption of corn starch.

The impact of starch concentration on the pesticide removal efficiency was investigated. Due to the noticeable settling down of starch granules over 2%, the impact of starch concentration was compared using 1% and 2% corn starch suspension (Figure 2C), which revealed that the 2% suspension was more effective in residue removal.

### 3.3. Optimizing the Homemade Soaking Strategy

Apples treated with the thiabendazole solution were submerged in various homemade aqueous systems (2% corn starch, 5% baking soda and the mixture) for 5 min. The mixture solution was composed of 2% corn starch and 5% baking soda. To be noted, a lower thiabendazole concentration (AI mass/area: 81.62 ng/mm^2^) was applied to the apples in the subsequent experiments to keep the residue values within the evaluative range based on the SERS mapping method. As demonstrated in Figure 4A and Appendix A, the thiabendazole residues were significantly diminished by 2% corn starch (70.27%) and 5% baking soda (74.15%). However, the mixture proved less effective (47.01%), indicating a significant antagonistic interaction between the two household ingredients. It was known that corn starch removes thiabendazole residues through physisorption and chemisorption, while baking soda worked through alkaline hydrolysis of the wax layer [17,18]. Appendix A illustrated the stability of suspensions obtained by Turbiscan, which indicated the baking soda accelerated the process of sedimentation. Additionally, the presence of baking soda increased the zeta potential from −18.533 mV to −7.286 mV, which was likely attributed to the adsorption of sodium ions (Na^+^) onto the starch particle surfaces. It neutralized some of the negative charges and thus promoted aggregation and sedimentation. Therefore, the reduced pesticide removal efficiency using the mixture compared to corn starch alone was due to the insufficient starch absorption of thiabendazole, attributed to reduced starch particles suspended in water. In addition, the diminished pesticide removal efficiency using the mixture compared to baking soda alone resulted from the reduced hydroxide ions being available for the hydrolysis of apple wax (Equation (1)), as the pH value of baking soda (5%) decreased from 8.8 to 8.6, which was likely due to the presence of natural acid in corn.(1)$${\mathrm{RCOOR}\prime +\mathrm{OH}}^{-}\to {\mathrm{RCOO}}^{-}+R\prime \mathrm{OH}$$
where RCOOR′ represents the wax ester, and RCOO^−^ represents the formed fatty acid anion.

The impact of soaking time on the thiabendazole removal efficiency using 2% corn starch and 5% baking soda was investigated. As shown in Figure 4B, no significant change in residue values was observed with increased soaking time for both washing agents, suggesting that extending the soaking time might not enhance the removal efficiency.

To further improve washing efficiency, apples were sequentially immersed in either agent (2% corn starch or 5% baking soda) for 5 min each. Additionally, the effectiveness of these soaking strategies was assessed using a more complex pesticide formulation, thiabendazole with Alligare 90^®^ (AI mass/area: 81.62 ng/mm^2^). Figure 5 and Appendix A revealed that the soaking strategy, 2% corn starch followed by 5% baking soda, significantly eliminated the residue values to 4.79 ± 2.47 ng/mm^2^ for T and 6.71 ± 2.25 ng/mm^2^ for TA, with thiabendazole reduced by 94.13% and 91.78%, respectively. However, reversing the soaking order was less effective in washing off T (74.48%) but yielded a comparable reduction for TA (81.42%), which was close to the removal efficiency in 5% baking soda alone for 10 min (79.46% for T and 80.27% for TA). These outcomes were likely attributed to the different mechanisms by which corn starch and baking soda remove thiabendazole from apples. Corn starch primarily eliminated the thiabendazole residues that stayed on apple surfaces through physisorption and chemisorption, while baking soda removed the alkali-resistant thiabendazole that was either embedded or interacted with the apples’ wax layer through alkaline hydrolysis of the wax [17,18]. The sequence of applying corn starch before baking soda effectively cleaned surface residues and subsequently dissolved the wax layer, leading to a more complete residue removal. In contrast, the opposite soaking sequence dissolved the wax layer, potentially limiting the sorption ability of corn starch. Soaking in baking soda alone could not remove surface residues evenly due to the wax crystal structure [53,54,55], while utilizing corn starch alone was less effective as it failed to address residues that were embedded in or that interacted with the wax layer. Therefore, sequential soaking using 2% corn starch followed by 5% baking soda is the most effective homemade soaking strategy for removing thiabendazole from apples.

### 3.4. Optimizing the Washing Strategy Using Commercial Agents

Similarly, to develop the most effective washing strategy using commercially available wash products, the residue values (mass/area) and corresponding reduction rate (%) of two thiabendazole-based formulations (AI mass/area: 81.62 ng/mm^2^) on apples treated with different washing (spraying or soaking) strategies were calculated using the SERS mapping method. To ensure a fair comparison with previously developed homemade soaking strategies, the commercial instructions were modified to exclude scrubbing (Appendix A). Among them, Product 1 and Product 4 were applied by spraying on the apple surfaces and stood for 15 min before water rising, while Product 2 and Product 3 were diluted and utilized as a soaking solution before rinsing. As the instructions for washing fresh produce varied by the product, a standardized soaking protocol was developed that entailed submerging apples in the 2% dilution of each product for 5 min followed by water rinsing for 30 s. To be noted, the manufacturer’s instruction for washing using Product 3 was the same as the soaking protocol.

The SERS analysis results for T and TA following different washing instructions and soaking protocols were summarized in Appendix A and Appendix A, respectively. When following the product instructions, all the products presented similar thiabendazole removal efficiencies, with reduction rates ranging from 50% to 73% in Appendix A. Figure 6A highlighted the outstanding performance of the soaking protocol with Product 4 (2%), which reduced the residue values to 3.84 ± 1.73 ng/mm^2^ for T and 3.52 ± 0.19 ng/mm^2^ for TA, with reduction rates of 95.3% for T and 95.69% for TA, respectively (Appendix A). The thiabendazole removal efficiency of soaking in Product 4 was close to that of the developed homemade washing strategy, with no significant difference in reduction rates for T (*p* = 0.597 > 0.05) and TA (*p* = 0.134 > 0.05). Figure 6B revealed that extending the soaking time did not further improve removal efficiency, which suggested that most of the pesticides had been removed within 5 min, with the remaining residues persistent due to their interactions with apple surfaces. Therefore, soaking in the 2% dilution of Product 4 for 5 min seemed to be the most effective washing strategy using commercial agents.

The comparative analysis (Figure 6A) indicated that soaking was generally more effective than spraying for pesticide removal, as it allowed for more sufficient interaction between the washing ingredients and pesticide residues. Additionally, there were no significant differences between the thiabendazole residues in the absence and presence of Alligare 90^®^ under each washing scenario.

The cleaning effect of soaking in Product 4 was significantly superior to that of the other products, which was likely due to the emulsifier ingredients, oleic acid and potassium oleate, which served as an effective cleansing agent (Appendix A). Although most of the ingredients, including these emulsifiers, are FDA-approved for use in Food (Title 21 CFR) [56], safety concerns arise with the surfactants, decyl glucoside, only approved for pre- and post-harvest agricultural practice (Title 40 CFR) [57], and lauryl glucoside, which lacks approval for use in food or agricultural practices. Therefore, despite its effectiveness, the use of Product 4 may not be as safe as the homemade soaking strategy.

### 3.5. Validating the Efficiency of Washing Strategies Using Homemade and Commercial Agents on Different FRUITS

LC-MS/MS analysis was conducted to determine the total thiabendazole residue in apples treated with TA (AI: 20 µg/g) followed by washing using the most effective homemade soaking strategy (sequential soaking in 2% corn starch and 5% baking soda for 5 min each) and the soaking strategy using the commercial agent (soaking in 2% Product 4 for 5 min). Figure 7 showed that the homemade soaking strategy removed 83.5% of thiabendazole residues from apples, 98.89% from grapes, 95.9% from lemons, and 88.7% from strawberries, which were comparable to the efficiency of soaking in a commercial agent (86% for apples, 97.9% for grapes, 97.5% for lemons, and 87.9% for strawberries). The ANOVA analysis results indicated that thiabendazole residues were more challenging to remove from apples and strawberries rather than grapes and lemons. The LC-MS/MS analyses confirmed that both washing strategies were effective in cleaning pesticide residues from various fruits. However, significant color fading and texture softening were observed on strawberries subjected to 5% baking soda. Hence, the homemade soaking strategy has a superior safety profile and greater accessibility, while the soaking protocol using Product 4 requires less washing time without compromising the quality of the fruits.

## 4. Conclusions

This work evaluated the efficiency of various washing strategies using either household ingredients or commercial agents for the removal of thiabendazole residues from fruits. Among the starches tested, corn starch (2%) exhibited the highest efficacy in eliminating thiabendazole residues from apples, primarily due to the physical entrapment of pesticide molecules and the formation of hydrogen bonds with thiabendazole. While the use of corn starch, baking soda, and their mixture showed limited removal capabilities, the sequential soaking protocol of 2% corn starch followed by 5% baking soda significantly enhanced the washing efficiency by reducing 94.13% and 91.78% of thiabendazole with and without Alligare 90^®^, respectively. This method was deemed as the most effective homemade washing strategy. On the other hand, the commercial wash agents were more effective when diluted for soaking apples rather than directly utilized for spraying on apple surfaces. Soaking in a 2% dilution of Product 4 removed 95.3% and 95.99% of thiabendazole with and without Alligare 90^®^ in apples and was positioned as the preferred commercial agent-based washing strategy. Both washing strategies using household and commercial agents showed similar thiabendazole removal efficiencies in apples, validated by SERS and LC-MS/MS analyses. These results also suggested that the non-ionic surfactant had no impact on pesticide removal. Furthermore, LC-MS/MS analysis revealed that both strategies were highly effective in a variety of fruits, including apples, grapes, lemons, and strawberries, without significant differences in efficacy. However, the washing strategy based on Product 4 was less preferable due to safety concerns associated with its surfactant ingredients. This study’s findings provide valuable instructions for effectively washing off pesticide residues from fruits, thereby contributing to food safety.

## Figures and Tables

**Figure 1 foods-14-00318-f001:**
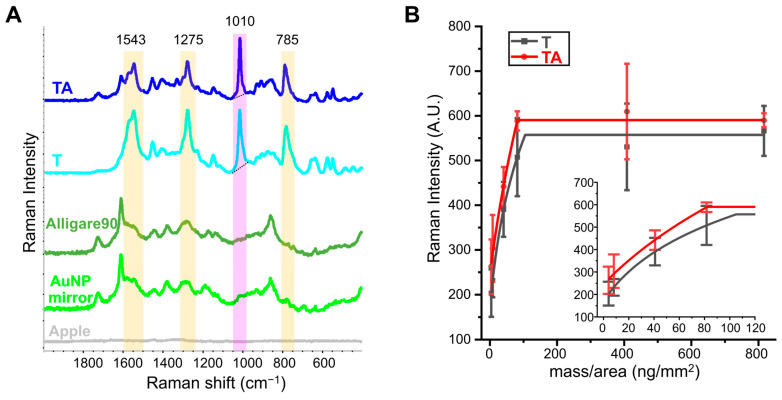
(**A**) Representative SERS spectra of TA (thiabendazole with Alligare 90^®^), T (thiabendazole alone), and Alligare 90^®^ using AuNP mirror substrates on apple surfaces. SERS spectra of the AuNP mirror and apple are shown as controls. (**B**) The stepwise regression curves of two thiabendazole-based solutions.

**Figure 2 foods-14-00318-f002:**
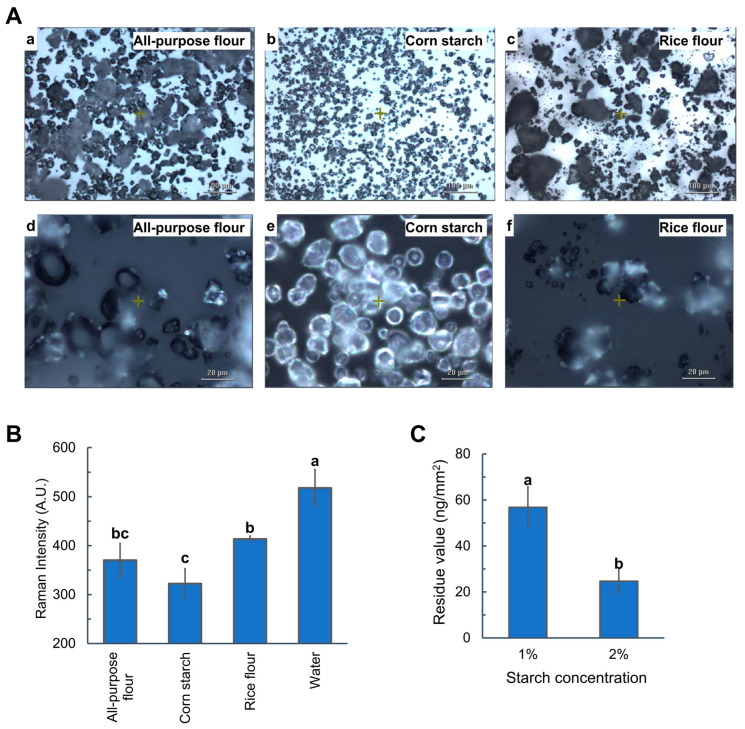
(**A**) Bright field images of all-purpose flour, corn starch, and rice flour granules under (**a**–**c**) 10× and (**d**–**f**) 50× objective lens. (**B**) Effect of suspension type on the Raman signal intensity at 1010 cm^−1^ and (**C**) effect of corn starch concentration on the thiabendazole residue values on the apple following 10 min soaking. Different letters in (**B**,**C**) indicate statistically significant differences (*p* < 0.05).

**Figure 3 foods-14-00318-f003:**
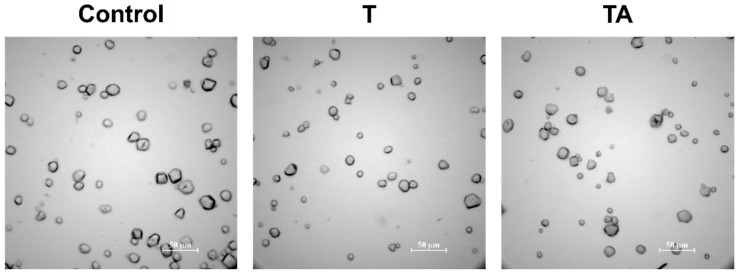
Bright-field microscopy images of corn starch suspension utilized for soaking apples without thiabendazole application (control), applied with T (thiabendazole alone) and applied with TA (thiabendazole with Alligare 90^®^).

**Figure 4 foods-14-00318-f004:**
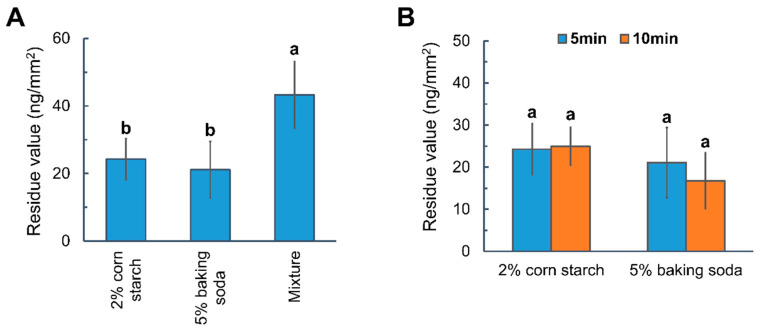
(**A**) Thiabendazole residues on apple following 5 min soaking using 2% of corn starch, 5% baking soda, and their mixture. (**B**) Effects of soaking time on the thiabendazole residue values following the strategy using corn starch and baking soda. Different letters indicate statistically significant differences (*p* < 0.05).

**Figure 5 foods-14-00318-f005:**
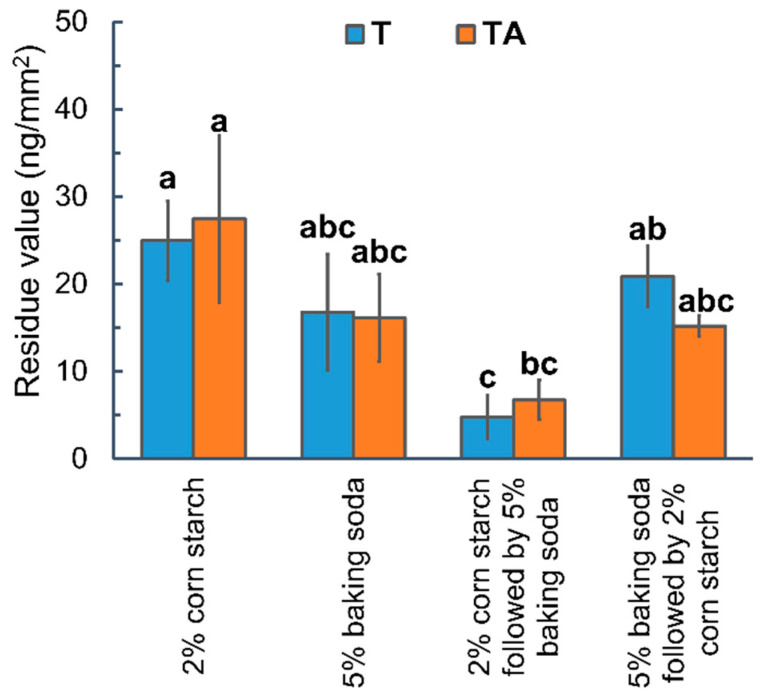
Thiabendazole residues on apple following different homemade soaking strategies. Different letters indicate statistically significant differences (*p* < 0.05).

**Figure 6 foods-14-00318-f006:**
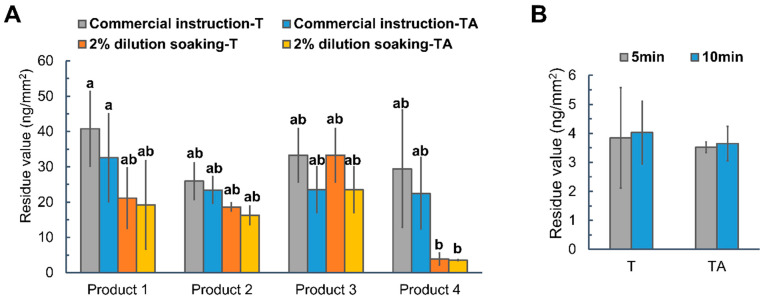
Thiabendazole residues of T (thiabendazole alone) and TA (thiabendazole with Alligare 90^®^) on apple (**A**) following varied washing strategies using commercial products and (**B**) following soaking strategy using 2% of Product 4 for different soaking time. Different letters indicate statistically significant differences (*p* < 0.05).

**Figure 7 foods-14-00318-f007:**
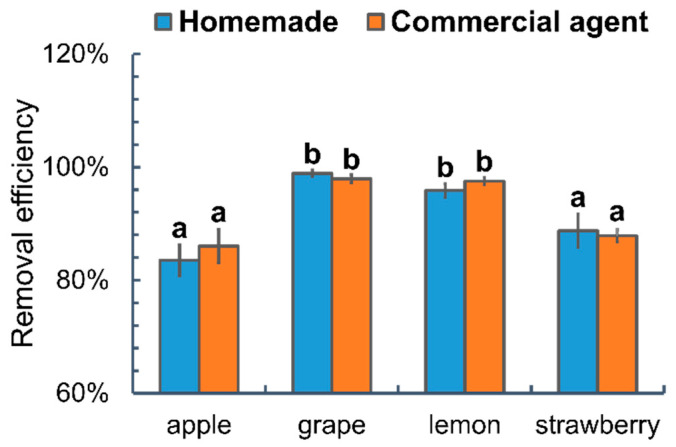
Analysis of thiabendazole residues in various fruits following the most effective homemade soaking strategy and soaking protocol using commercial agent using LC-MS/MS. Different letters indicate statistically significant differences (*p* < 0.05).

**Table 1 foods-14-00318-t001:** Regression equations with corresponding residue value range and determination coefficient (R^2^) for two thiabendazole-based solutions on apples.

	Regression Equation	Range (ng/mm^2^)	R^2^
T	y = −10.80 + 56.01x	0–4.08	0.958
	y = −722.96 + 260.56 ln(x + 30.86)	4.08–105.23	0.992
	y = 557.25	105.23–816.25	
TA	y = 7.59 + 61.17x	0–4.08	0.997
	y = −4241.88 + 867.40 ln(x + 180.51)	4.08–82.15	0.999
	y = 590.28	82.15–816.25	

## Data Availability

The original contributions presented in this study are included in the article/Appendix A. The raw data supporting the conclusions of this article will be made available by the authors on request.

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
