# Peer review of "Efficacy of Household and Commercial Washing Agents in Removing the Pesticide Thiabendazole Residues from Fruits"

_foods, 2025, doi:10.3390/foods14020318_

Round 1

Reviewer 1 Report

Comments and Suggestions for Authors

Translator        

This study investigates the efficacy of household ingredients (corn starch, all-purpose flour, rice flour and baking soda) and four commercial fresh produce wash products in eliminating a model pesticide thiabendazole with and without a model non-ionic surfactant Alligare 90® from postharvest fruits. Surface-enhanced Raman spectroscopy (SERS) was employed to swiftly quantify residual thiabendazole on apple surfaces following treatment with these cleaning agents. From the perspectives of residue levels and safety assurance, this study offers critical guidance for the effective removal of pesticide residues from fruits, thereby ensuring enhanced food safety. The manuscript's concept is clearly articulated, featuring a sound experimental design supported by adequate data.  There are following points that the authors should address first.

1. Please check the statements on lines 299-301, such as table2.

2. Please add the spectra and calibration curves verified by LC-MS/MS.

3. Please check and standardise the format of references, e.g. journal names are either abbreviated or in full.

4. The author examined the cleaning efficacy of both household and commercial detergents, alongside analyzing the residual substances left behind post-washing. Does the author tend to choose one detergent over the other? Alternatively, does the study suggest that different fruits and types of pesticide residues require the selection of specific detergents for optimal results?

Comments on the Quality of English Language

Translator        

 The English could be improved.

Author Response

Comments 1: Please check the statements on lines 299-301, such as table2.

Response 1: Thank you very much for pointing it out. The content in Line 299-301 were written as: “The impact of starch concentration on the pesticide removal efficiency was investigated. Due to the noticeable settling down of starch granules over 2%, the impact of starch concentration was compared using 1% and 2% corn starch suspension (Figure 2C), which revealed that the 2% suspension was more effective in residue removal.”

Comments 2: Please add the spectra and calibration curves verified by LC-MS/MS.

Response 2: Thank you for your suggestions. Surface-enhanced Raman spectroscopy (SERS) has been widely considered as a rapid and reliable method for the quantitative analysis of pesticide residues directly on fresh produce surfaces [1–4]. In this study, the residues were measured on fruits that had only been exposure to pesticide for one hour, which is insufficient for significant pesticide penetration. Under these conditions, SERS can provide accurate residue quantification, eliminating the need for additional LC-MS/MS confirmation for calibration.

Comments 3: Please check and standardize the format of references, e.g. journal names are either abbreviated or in full.  

Response 3: Thank you for your suggestions. We have checked all the references and ensured all the journal articles have proper journal names.

Comments 4: The author examined the cleaning efficacy of both household and commercial detergents, alongside analyzing the residual substances left behind post-washing. Does the author tend to choose one detergent over the other? Alternatively, does the study suggest that different fruits and types of pesticide residues require the selection of specific detergents for optimal results?

Response 4: This study demonstrated comparable washing efficiency between the sequential soaking protocol (2% corn starch followed by 5% baking soda) and soaking in a 2% solution of commercial Product 4 across various fruits. However, as noted in Page 11: “Although most of the ingredients, including these emulsifiers, are FDA-approved for use in Food (Title 21 CFR) [52], safety concerns arise with the surfactants, decyl glucoside, only approved for pre- and post-harvest agricultural practice (Title 40 CFR) [53], and lauryl glucoside, which lacks approval for use in food or agricultural practices. Therefore, despite its effectiveness, the use of Product 4 may not be safe as the homemade soaking strategy.” In a word, the sequential soaking protocol (2% corn starch followed by 5% baking soda) should be more recommended.

Reference:

  1. Tang, J.; Zhang, Q.; Zhou, J.; Fang, H.; Yang, H.; Wang, F. Investigation of Pesticide Residue Removal Effect of Gelatinized Starch Using Surface-Enhanced Raman Scattering Mapping. Food Chem 2021, 365, 130448, doi:10.1016/J.FOODCHEM.2021.130448.
  2. Ye, C.; He, M.; Zhu, Z.; Shi, X.; Zhang, M.; Bao, Z.; Huang, Y.; Jiang, C.; Li, J.; Wu, Y. A Portable SERS Sensing Platform for the Multiplex Identification and Quantification of Pesticide Residues on Plant Leaves. J Mater Chem C Mater 2022, 10, 12966–12974, doi:10.1039/D2TC02926J.
  3. Chen, J.; Dong, D.; Ye, S. Detection of Pesticide Residue Distribution on Fruit Surfaces Using Surface-Enhanced Raman Spectroscopy Imaging. RSC Adv 2018, 8, 4726–4730, doi:10.1039/C7RA11927E.
  4. Du, X.; Gao, Z.; He, L. Quantifying the Effect of Non‐ionic Surfactant Alkylphenol Ethoxylates on the Persistence of Thiabendazole on Fresh Produce Surface. J Sci Food Agric 2024, 104, 2630–2640, doi:10.1002/jsfa.13147.

Reviewer 2 Report

Comments and Suggestions for Authors

The topics of the manuscript are in line with the trends of today's science. Methods are being sought to effectively remove pesticides from fruit, vegetables or cereals.

Below are a few comments:

Why was thiabendazole chosen for the study?

In the introduction, provide information about its properties and effects on organisms.

Why was such a fruit chosen for the study?

Line 188 - briefly describe the method of preparing the sample for testing.

Item 39 reference (line 188, 212) If the work has not been published, how can the reader refer to it for details?

No discussion with other researchers. Complete this, please.

Why was the SERS technique used to determine only thiabendazole residues in apples and the LC-MS/MS technique in all fruit?

Author Response

Comments 1: Why was thiabendazole chosen for the study?

Response 1: Thiabendazole was chosen for this study because it is widely utilized as a fungicide in agriculture, particularly for post-harvest treatments to control fungal diseases in fruits and vegetables [1]. Our previous research on the persistence of various pesticide residues across different fresh produce matrices revealed that thiabendazole is particularly challenging to fully remove [2–4], making it an ideal model pesticide for evaluating the efficacy of washing strategies.

Comments 2: In the introduction, provide information about its properties and effects on organisms.

Response 2: Thank you for your suggestions. Information regarding the properties and effects of thiabendazole is provided in Line 56: “Thiabendazole, a systemic fungicide approved for postharvest application on various fruits, was selected as the model pesticide. Although thiabendazole is generally considered low in acute toxicity to mammals (oral LD50 for rats: 2080 mg/kg) [23], it poses potential developmental toxicity risks [24]. Furthermore, its persistence on fresh produce has raised significant food safety concerns [25,26]. The maximum residue limit (MRL) of thiabendazole set by Environmental Protection Agency (EPA) is 10 µg/g, 0.05 µg/g, 10 µg/g and 5 µg/g for apple, grape, lemon and strawberry, respectively [27].”

Comments 3: Why was such a fruit chosen for the study?

Response 3: Thank you for your question. The fruit types having the highest frequency of thiabendazole residue issues were chosen based on USDA 2021 annual report [5].

Comments 4: Line 188 - briefly describe the method of preparing the sample for testing.

Response 4: Thank you for your comment. The extraction details follow the standard QuEChERS operating procedure for pesticide residue analysis [6]. Additional description was not provided in the manuscript, as the QuEChERS operating procedure is widely recognized and referenced in the manuscript. However, if further clarification is needed, we would be happy to expand on this description in the revised manuscript.

Comments 5: Item 39 reference (line 188, 212) If the work has not been published, how can the reader refer to it for details? No discussion with other researchers. Complete this, please.

Response 5: Thank you for pointing it out. The paper has now been officially accepted, and the citation has been updated as follows:  Du, X.; Doherty, J.; Lee, J.; Clark, J.M.; He, L. Assessment of the Effect of Non-Ionic Surfactant Alkylphenol Ethoxylates on the Penetration of Pesticides in Fresh Produce. Spectrochim Acta A Mol Biomol Spectrosc 2024, 125691, doi:10.1016/j.saa.2024.125691.

Comments 6: Why was the SERS technique used to determine only thiabendazole residues in apples and the LC-MS/MS technique in all fruit?

Response 6: We previously developed and validated a reliable, rapid, and cost-effective SERS method and standard curve for apples [7]. This allowed us to efficiently evaluate the effects of different washing agents. For other fruits, no SERS method has been previously established or validated, so LC-MS/MS was used for final validation tests. Given the high cost of LC-MS/MS analysis ($175 per sample), its use was minimized to essential validation tests.  

Reference:

  1. 421. Thiabendazole (Pesticide Residues in Food: 1977 Evaluations) Available online: https://www.inchem.org/documents/jmpr/jmpmono/v077pr43.htm (accessed on 3 January 2025).
  2. Yang, T.; Doherty, J.; Zhao, B.; Kinchla, A.J.; Clark, J.M.; He, L. Effectiveness of Commercial and Homemade Washing Agents in Removing Pesticide Residues on and in Apples. J Agric Food Chem 2017, 65, 9744–9752, doi:10.1021/acs.jafc.7b03118.
  3. Yang, T.; Zhao, B.; Hou, R.; Zhang, Z.; Kinchla, A.J.; Clark, J.M.; He, L. Evaluation of the Penetration of Multiple Classes of Pesticides in Fresh Produce Using Surface‐Enhanced Raman Scattering Mapping. J Food Sci 2016, 81, T2891–T2901, doi:10.1111/1750-3841.13520.
  4. Du, X.; Gao, Z.; Yang, T.; Qu, Y.; He, L. Understanding the Impact of a Non-Ionic Surfactant Alkylphenol Ethoxylate on Surface-Enhanced Raman Spectroscopic Analysis of Pesticides on Apple Surfaces. Spectrochim Acta A Mol Biomol Spectrosc 2023, 301, 122954, doi:10.1016/j.saa.2023.122954.
  5. PDP 2021 Annual Summary Available online: http://www.fda.gov/Food/Chemicals-Metals-Pesticides- (accessed on 4 January 2025).
  6. Lehotay, S.J.; O’Neil, M.; Tully, J.; García, A.V.; Contreras, M.; Mol, H.; Heinke, V.; Anspach, T.; Lach, G.; Fussell, R.; et al. Determination of Pesticide Residues in Foods by Acetonitrile Extraction and Partitioning with Magnesium Sulfate: Collaborative Study. J AOAC Int 2007, 90, 485–520, doi:10.1093/JAOAC/90.2.485.
  7. Du, X.; Gao, Z.; He, L. Quantifying the Effect of Non‐ionic Surfactant Alkylphenol Ethoxylates on the Persistence of Thiabendazole on Fresh Produce Surface. J Sci Food Agric 2024, 104, 2630–2640, doi:10.1002/jsfa.13147.

Round 2

Reviewer 2 Report

Comments and Suggestions for Authors

I have no comments.